# Genetic Interference of FGFR3 Impedes Invasion of Upper Tract Urothelial Carcinoma Cells by Alleviating RAS/MAPK Signal Activity

**DOI:** 10.3390/ijms24021776

**Published:** 2023-01-16

**Authors:** Gong-Kai Huang, Chao-Cheng Huang, Chih-Hsiung Kang, Yuan-Tso Cheng, Po-Ching Tsai, Ying-Hsien Kao, Yueh-Hua Chung

**Affiliations:** 1Department of Pathology, Kaohsiung Chang Gung Memorial Hospital and Chang Gung University College of Medicine, Kaohsiung 83301, Taiwan; 2Biobank and Tissue Bank, Kaohsiung Chang Gung Memorial Hospital, Kaohsiung 83301, Taiwan; 3Department of Urology, Kaohsiung Chang Gung Memorial Hospital and Chang Gung University College of Medicine, Kaohsiung 83301, Taiwan; 4Department of Medical Research, E-Da Hospital, I-Shou University, Kaohsiung 82445, Taiwan; 5Department of Medical Laboratory Science and Biotechnology, Kaohsiung Medical University, Kaohsiung 80708, Taiwan

**Keywords:** UTUC, FGFR3, epithelial–mesenchymal transition, RAS/MAPK signaling

## Abstract

Upper tract urothelial cancer (UTUC) is a less common disease in Western countries but has a high level of prevalence in Asian populations. Compared to bladder cancer, unique etiologic and genomic factors are involved in UTUC. Fibroblast growth factor receptor 3 (FGFR3) up-regulation has been proposed as a promising target for bladder cancer therapy. In this study, we aimed to profile the expression of FGFR3 in Asian and Caucasian UTUC tissues and to evaluate the in vitro therapeutic efficacy of small interference RNA (siRNA)-mediated FGFR3 silencing in UTUC treatment. The FGFR3 expression levels in renal pelvis tissues and microarray sections from Asian and Caucasian patients with UTUC, respectively, were measured via immunohistochemistry. The BFTC-909 and UM-UC-14 UTUC cell lines were used to examine the effects of FGFR3 silencing on proliferation, migration, epithelial–mesenchymal transition (EMT) marker expression, and signaling machinery. FGFR3 expression increased as the TNM stage increased in both Asian and Caucasian UTUC tumors, and no statistical difference was identified between the two groups. In vitro studies demonstrated that FGFR3 siRNA delivery significantly inhibited proliferation and migration and suppressed the expression of EMT markers and transcription factors in UTUC cells. Mechanistically, FGFR3 silencing alleviated the constitutive expression of RAS and the phosphorylation of MAPK signaling mediators, including ERK1/2 and JNK1/2. FGFR3 silencing elicited an apoptosis-inducing effect similar to that of FGFR inhibition. Conclusion: siRNA-targeted FGFR3 expression may impede the expansion and invasion of UTUC cells by alleviating the RAS/MAPK signaling pathway. The genetic interference of FGFR3 expression via siRNA in UTUC cells may constitute a useful therapeutic strategy.

## 1. Introduction

Urothelial carcinoma (UC) is a common cancer that occurs in the epithelium lining of the urinary tract, including bladder cancer, ureteral carcinoma, and renal pelvis UC. Despite being a relatively uncommon malignancy in Western countries, upper tract urothelial cancer (UTUC), including ureter and renal pelvis UC, represents more than 40% of UC cases in Taiwan [1,2,3,4]. The higher level of prevalence of UTUC in Asian populations indicates that unique etiologic and genomic factors, distinct from bladder cancer, may contribute to UTUC tumorigenesis [5]. In light of the poor therapeutic outcomes of advanced UTUC [1], diagnostic tools and novel therapeutic strategies for UTUC, treatment must urgently be developed.

Epithelial–mesenchymal transition (EMT) is a cellular process through which epithelial cells are able to reduce cell-to-cell contact and gain mesenchymal characteristics, thereby increasing cell migration during diverse cellular events, including embryogenesis and tumorigenesis [6]. Epithelial cells undergoing the EMT process may display the decreased expression of epithelial marker E-cadherin and conversely show the increased expression of transcription factors and mesenchymal markers, including vimentin, α-smooth muscle actin, and fibronectin. The increased expression of EMT markers has been found in UTUC tissues, and EMT markers are regarded as significant diagnostic and prognostic biomarkers [7,8,9,10,11]. Meanwhile, EMT induction with the concurrent degradation of the extracellular matrix (ECM), which facilitates tumor metastasis, is essential for the migration and invasion of cancer cells [9,10,11,12,13,14]. In this regard, matrix metalloproteinases (MMPs), including MMP9, are key enzymes for ECM degradation [15], while intratumoral MMP9 expression is positively correlated with the aggressiveness of many cancers [14,15,16,17,18]. In terms of signaling, mitogen-activated protein kinase (MAPK)-mediated signaling mechanistically contributes to EMT induction and MMP9 activation [15,19]. Indeed, blockades in the EMT process and MMP activation have been considered plausible therapeutic targets for tumor treatment [14,16].

Fibroblast growth factor receptor 3 (FGFR3) is a member of the FGFR family, which belongs to a subfamily of receptor tyrosine kinases, mediates signal transduction, and is known to regulate a diversity of physiological and pathological processes, including cell proliferation, differentiation, and tumorigenesis [20,21,22]. The FGFR3-activated signaling cascades, including the RAS-dependent MAPK pathway, may promote tumorigenesis [23,24]. Accumulating evidence indicates that gain-of-function mutations in the *FGFR3* gene may give rise to constitutive receptor activation and have been identified in bladder cancer and UTUC [25,26,27,28,29]. Accordingly, the FGFR3/RAS/MAPK signaling axis has long been proposed as a target for UC treatment [5,30]. Despite intensive discussion regarding the application of targeting FGFR3 using receptor kinase inhibitors or ligand-trapping antibodies in UC treatment [31,32,33], little is known about the efficacy of the FGFR3 genetic interference strategy in UTUC treatment. To approach the application of FGFR3 targeting, in this study, we aimed to profile and compare the expression patterns of FGFR3 between Asian and Caucasian UTUC tissue specimens to explore the in vitro efficacy of the small interference RNA (siRNA)-mediated *FGFR3* gene silencing strategy, and to elucidate possible underlying molecular mechanisms.

## 2. Results

### 2.1. FGFR3 Expression Was Up-Regulated in Human Renal Pelvis UTUC Tissues

To observe the expression and distribution of FGFR3 in human renal pelvis UTUC tissues, a TMA mainly collecting formalin-fixed paraffin-embedded sections from Caucasian patients with UTUC was subjected to IHC staining. The microscopic observation clearly showed that FGFR3 expression was seen in the cytoplasm of most tumor cells at all stages (Figure 1A), and the FGFR3 antigenic intensity significantly increased along with the malignancy of UTUC (Figure 1B). Negative to moderate FGFR3 IHC-staining intensities were found in the kidney tissues at stages T1–2, while moderate to intense FGFR3 signals were detected in all the Caucasian specimens at the advanced T3 and T4 stages (Table 1). Notably, the histological antigenicity of FGFR3 in renal pelvis tissues was heterogeneously distributed in proximal and distal tubular epithelia, i.e., proximal renal tubules usually showing weak FGFR3 expression (score 1+), while distal tubules in the same specimen mostly displayed stronger FGFR3 signals (score 2+~3+) (Figure 1C). Similar to TMA data, increased FGFR3 expression was also noted in the UTUC tissues collected from Asian patients (Figure 2A,B). High FGFR3 positivity rates were observed in normal renal pelvis tissues, which displayed scores of 1+~2+, whereas 15.8% and 10% of Asian patients with UTUC at stages T3 and T4, respectively, showed FGFR3-negative results (Table 2). Moreover, the statistical analysis between these two ethnic groups indicated that only Caucasian patients with UTUC at stage T4 prominently expressed higher FGFR3 intensity levels than Asian patients (*p* = 0.039, Figure 2C). Taken together, these data support the oncogenic role of up-regulated FGFR3 expression in the development of renal pelvis tumors in both Asian and Caucasian populations.

### 2.2. FGFR3 siRNA Delivery Reduced Proliferation and Migration of Cultured UTUC Cells

To investigate the effects of *FGFR3* gene depletion on the proliferation and migration of UTUC cells, FGFR3 siRNA was transfected into BFTC-909 and UM-UC-14 cells, both of which are widely used in UTUC studies [2,9,11,34]. A water-soluble WST-1 tetrazolium cell proliferation assay was used to monitor tumor cell growth for 2 or 3 consecutive days. The assay results clearly showed that FGFR3 siRNA delivery completely abrogated the growth of both BFTC-909 (Figure 3A) and UM-UC-14 UTUC cells (Figure 3B).

To observe the effect of *FGFR3* gene silencing on tumor cell migration, UTUC tumor cells were subjected to a wound-healing migration assay. Microscopic observations on the closure of the mimicked wound area clearly showed that FGFR3 siRNA delivery markedly suppressed the motility of both BFTC-909 (Figure 4A) and UM-UC-14 UTUC cells (Figure 4B). The suppressive effects of FGFR3 siRNA treatment on the growth and migration of UTUC cells indicate that FGFR3 could serve as a target for UTUC treatment.

### 2.3. FGFR3 Gene Silencing Suppressed Expression of EMT Markers in Cultured UTUC Cells

To investigate the involvement of FGFR3 expression in UTUC cells in the regulation of EMT processes, BFTC-909 cells derived from an Asian patient were selected to measure the cellular expression of EMT markers and transcription factors in the FGFR3 gene-silenced cells by using IF staining. The analysis of the IF signal intensities clearly showed that FGFR3 siRNA delivery significantly increased the cellular expression of an epithelial marker, E-cadherin (Figure 5A,D), and simultaneously decreased that of mesenchymal cell markers, including vimentin, MMP9, fibronectin (Figure 5A,B,D), and that of an EMT transcription factor, Snail protein (Figure 5C,D). Moreover, the results of our previous study demonstrated that the increased expression of claudin-5, a tight junction structural protein, may reflect the inhibition of EMT processes in UTUC cells [9]. It was shown that FGFR3 siRNA delivery also significantly increased the claudin-5 expression in BFTC-909 UTUC cells (Figure 5C,D). Altogether, these data strongly suggest that siRNA-mediated *FGFR3* gene knockdown might suppress tumor metastasis by blocking the EMT machinery and relevant signaling activity in the UTUC cells.

### 2.4. FGFR3 Gene Silencing Alleviated RAS/MAPK Signaling Cascade and Induced Apoptosis in Cultured UTUC Cells

Given the involvement of FGFR3-driven and/or RAS/MAPK signaling activation in tumorigenesis, including UTUC development [19,23], we next observed the molecular alterations in the UTUC cells receiving FGFR3 siRNA or scramble non-target control transfection. The Western blotting data clearly showed that 48 h of FGFR3 siRNA delivery effectively knocked down its cellular contents in the BFTC-909 UTUC cells (Figure 6A,B). The FGFR3 silencing treatment simultaneously resulted in the significant down-regulation of constitutive RAS expression (Figure 6A,C), as well as the hypophosphorylation of MAPK signaling mediators, including p44/42 ERK1/2 (Figure 6A,D) and p54/46 JNK1/2 (Figure 6A,E) in the BFTC-909 cells. Furthermore, a cell proliferation assay and TUNEL staining were used to examine the cytotoxic and apoptogenic effects of siRNA delivery and to compare them with those of receptor kinase inhibition. The morphologic observation and cell proliferation assay clearly showed that FGFR3 silencing apparently exhibited lower cytotoxicity compared with PD173074, an FGFR inhibitor acting on both FGFR1 and FGFR3 (cell morphology shown in Appendix A). Meanwhile, FGFR3 silencing elicited the prominent apoptosis of BFTC-909 UTUC cells, although the apoptotic induction rate was not as high as that induced by PD173074 treatment (Figure 7A,B). Altogether, these results demonstrate that FGFR3 siRNA delivery can significantly suppress the proliferation, migration, and expression of EMT markers and transcription factors in cultured UTUC cells, and they indicate that the suppressive effects of *FGFR3* gene silencing are most likely mediated through alleviation of the RAS/MAPK signaling cascade (the proposed mechanism is depicted in Figure 8).

## 3. Discussion

To our knowledge, this study is the first to profile and compare FGFR3 expression levels between Asian and Caucasian UTUC specimens. The IHC staining results indicated that the FGFR3 expression was up-regulated as the TMN/stages increased in the renal pelvis tissues of both Asian and Caucasian patients. Intriguingly, a statistical difference in FGFR3 expression rates between these two ethnic groups, i.e., Asian and Caucasian patients, was only noted in the patients with advanced UTUC at stage T4. It is worth mentioning that the number of tumor specimens at the advanced T4 stage collected from Caucasian patients (*n* = 5) differed from that of Asian patients (*n* = 30). We cannot rule out the possibility that the limited number of Caucasian specimens (i.e., from five patients) collected from advanced UTUC tumors might mean that the statistical significance was underestimated, while this is very likely due to the lower prevalence of UTUC in the Caucasian population. Overall, marked FGFR3 up-regulation could be seen in the development of renal pelvis tumors in both Asian and Caucasian populations. In addition, the unavailability of clinicopathological features from patients in TMA observation also limits the significance of the present study. Further study with the collection of more Caucasian specimens may confirm data validation. 

In addition to FGFR3 overexpression, FGFR3 mutations more frequently found in carcinoma in situ are believed to contribute to the development of superficial bladder cancer. In bladder UC, a low level of FGFR3 expression has been demonstrated to exhibit no association with low-grade, non-invasive lesions and a lower recurrence rate [26,35]. Given the mutual exclusion of FGFR3 and TP53 gene mutations in UC, the decreased frequency of FGFR3 mutations alongside the advancement of tumor stage and grade of dysplasia may be due to the fact that FGFR3-mutated tumors rarely progress, and most advanced cancers would derive from precursors with mutations in a different pathway for tumor development [26]. In contrast, FGFR3 genomic alterations may occur with a significantly higher frequency in UTUC [30,36,37]. Meanwhile, the overexpression of FGFR3 is considered to have muscle-invasive potential in bladder UC [38,39]. Accordingly, it has been claimed that the inhibition of FGFR signaling has clinical benefits for patients with genomic alterations in FGFRs [31]. In this regard, the clinical efficacy of FGFR-targeted therapies using selective FGFR kinase inhibitors in solid tumors, including UC, has recently emerged [32,40,41]. However, the low response rate may limit the development of FGFR-targeted therapy due to responses only occurring in patients with tumors harboring FGFR1-3 point mutations or fusions [32,41]. In light of the higher prevalence of UTUC in Asian countries such as Taiwan [1,2,3,4], taken with the significant up-regulation of FGFR3 noted in both Asian and Caucasian patients with advanced UTUC, we reasonably proposed that mutational differences in the *FGFR3* gene between Asian and Caucasian populations may be the cause and that the genetic interference of FGFR3 using siRNA may be an effective therapeutic strategy for UTUC treatment regardless of *FGFR3* gene mutations. However, the preclinical efficacy and safety of FGFR3 siRNA delivery require further investigation.

Another line of experience in immune checkpoint inhibitor therapy may support the notion that targeting FGFR3 has more therapeutic potential in Asian populations than in Caucasian populations. It is indeed believed that different epidemiology, genetic susceptibility, and molecular profiles may influence clinical benefits, including the response and survival rates in cancer therapies [30]. A recent meta-analysis study revealed that Asian cancer patients receiving PD-1/PD-L1 (programmed cell death 1)/(programmed death-ligand 1)-inhibitor-based therapy may experience a significantly improved benefit in terms of survival rate when compared to non-Asian patients [42]. Despite the fact that the ethnic differences between Asian and Caucasian populations in response to immune checkpoint inhibitor therapy in non-small-cell lung cancer remain controversial [43], it has been claimed that characteristics of Asian patients, including higher EGFR mutation rates, hepatitis B virus infection, and related immunotoxicity, are contributing factors [44]. In alignment with this notion, the higher efficacy and response toward combined immune checkpoint inhibitor treatment was observed in Taiwanese patients with advanced or metastatic UC in terms of prolonged progression-free survival and overall survival [45,46,47]. The same rationale may also underlie the different levels of effectiveness of FGFR3 kinase inhibitor therapies between Asian and Caucasian patients with UTUC.

In the context of signal transduction, FGFRs have been demonstrated to activate multiple pathways, including the RAS/MAPK cascade, which may initiate and promote tumor progression [48,49]. Previous mechanistic studies have validated the involvement of RAS/MAPK activities in the survival and EMT induction of UC cells [5,19,30]. To the best of our knowledge, in this study, we presented the first in vitro evidence showing that FGFR3 siRNA delivery significantly exhibits antitumor activity in terms of not only the suppression of proliferation and migration but also a reduction in the constitutive phosphorylation status of RAS/MAPK machinery in the UTUC cell line. Moreover, FGFR3 silencing suppressed the expression of Snail protein, an EMT transcription factor that has prognostic significance and implications for the invasion of UTUC tumors [10]. Our findings strongly suggest that the genetic depletion of FGFR3 may have therapeutic benefits in that, similar to receptor blockade and kinase inhibition, it can effectively shut down FGFR3-driven upstream signaling irrespective of gene mutations. Thus, we highly recommend FGFR3 genetic interference as an applicable therapeutic option for the treatment of patients with UTUC with tumors harboring *FGFR3* gene mutations and/or overexpression, as well as for those resistant to receptor kinase inhibitor treatment. Future preclinical studies regarding the efficacy of FGFR3 siRNA delivery may confirm its clinical applicability.

In conclusion, in this study, we reported FGFR3 overexpression in Asian and Caucasian specimens with advanced UTUC and demonstrated that, in addition to the inhibition of growth and migration, the genetic interference of FGFR3 via gene-specific siRNA delivery may suppress the expression of EMT markers and transcription factors in cultured UTUC cells. The molecular mechanism, at least in part, involves the alleviation of constitutive RAS/MAPK signal activity in tumor cells via *FGFR3* gene silencing. Our findings support the concept that the genetic interference of FGFR3 may constitute an applicable therapeutic strategy in UTUC treatment.

## 4. Materials and Methods

### 4.1. Human Kidney Cancer Tissues and Immunohistochemistry

Asian renal pelvis tissues were clinically collected from patients diagnosed with UTUC (*n* = 112) and archival normal kidney pelvis tissues (*n* = 50) at Kaohsiung Chang Gung Memorial Hospital. The study protocol was approved by the Institutional Review Board of Chang Gung Memorial Hospital (IRB no. 201801191B0) and complied with the Helsinki Declaration. In addition, a commercially available human kidney cancer tissue microarray (TMA) was purchased from US Biomax, Inc. (Kit no. HRaU-Uro120Sur-01, Rockville, MD, USA). The TMA specimens were mainly collected from Caucasian patients diagnosed with either kidney pelvis UC (*n* = 84) or ureteral UC (*n* = 35). The human kidney cancer TMA and UTUC tissue slides were subjected to immunohistochemistry (IHC) staining as previously described [8], using an anti-FGFR3 antibody (Cat. No. JM110-33, Invitrogen ThermoFisher Scientific, Waltham, MA, USA) and a polymer-based IHC detection kit (Cat. No. 87-8963, Zymed Laboratories, South San Francisco, CA, USA), followed by DAB chromogen visualization (Cat. No. K3467, DAKO, Carpinteria, CA, USA). The FGFR3 antigen intensity was quantified by pathologists and given a score from 1+ to 4+ according to the following rules: A score of 1+ indicates weak and granular cytoplasmic FGFR3 signals found in some tumor cells. A score of 2+ indicates moderate FGFR3 expression in most tumor cells. A score of 3+ indicates tumor cells displaying diffuse, strong, and dot-like FGFR3 signals in the cytoplasm. A score of 4+ means that samples display the strongest FGFR3 signals.

### 4.2. Cell Culture, FGFR3 siRNA Delivery, and Kinase Inhibitor Treatment

Two UTUC cell lines, clone BFTC-909 and UM-UC- 14, were cultured as described previously [9,11]. To observe the biomodulatory effect of *FGFR3* gene silencing on UTUC cell behaviors, the cells were transiently transfected with FGFR3 siRNA (Cat. No. S534558, ThermoFisher Scientific, Waltham, MA, USA) or scramble non-target control oligonucleotides at 10 nM by using Lipofectamine 2000 (Invitrogen ThermoFisher Scientific) following the manufacturer’s instructions for in vitro siRNA delivery. To compare the effect of *FGFR3* gene silencing with that of receptor kinase inhibition, the cells were treated with PD173074 (Cat. No. T2642, Targetmol, Wellesley Hills, MA, USA) at 10 nM.

### 4.3. Cell Proliferation Assay

The cell proliferation assay was performed by using a WST-1 assay kit (Cat. No. MK400, TaKaRa Bio Inc., Shiga, Japan) as previously described [8,9]. Cells grown on microtiter plates were treated with premixed WST-1 reagent for 4 h. The absorbance of a formazan end-product was measured at 440 nm and a reference wavelength at 600 nm using a microplate reader.

### 4.4. Wound-Healing Cell Migration Assay

The cell migration assay was performed using a wound-healing migration protocol as previously described [50]. The subconfluent monolayer of cells was seeded on dishes equipped with wound-mimicking inserts (Ibidi, Martinsried, Germany). The wound-mimicking inserts were removed after 24-h adherence. Afterward, FGFR3 siRNA and non-target control oligonucleotides were added, followed by daily image documentation under a phase contrast microscope. 

### 4.5. Western Blot Analysis

Cellular protein lysates were collected by using RIPA lysis buffer (Cell Signaling Technology, Billerica, MA, USA) premixed with protease and phosphatase inhibitors (ThermoFisher). Protein concentrations were measured using a protein assay dye (Bio-Rad Laboratories, Hercules, CA, USA). SDS-PAGE and immunoblotting analyses were performed as described previously [8,9,50]. The detecting antibodies used for immunodetection are listed in Appendix A. 

### 4.6. Immunofluorescent (IF) Staining

The UTUC cells were cultured in six-well glass slide chambers for 24 h and further treated with FGFR3 siRNA for 48 h. The cells were then fixed with 4% paraformaldehyde, permeabilized with 0.25% Triton X-100, and blocked with 3% BSA for 30 min at room temperature. The fixed cells were then incubated with the primary antibodies against E-cadherin, vimentin (2707-1 Epitomics, Burlingame, CA, USA), MMP9, fibronectin, claudin-5 (ab15106, Abcam, Waltham, MA, USA), and Snail (MA5-14801, ThermoFisher Scientific) at 4 °C overnight, followed by visualization with Alexa Fluor 488- or Alexa Fluor 595-conjugated secondary antibodies at room temperature for 1 h. The nuclei were counterstained with DAPI. The stained cells were mounted with a fluorescent mounting medium (Dako Cytomation, Glostrup, Denmark) and observed under fluorescence microscopy (Olympus, Tokyo, Japan). The exposure gains and rates were consistent between the samples. The fluorescent intensities were quantified on independent color channels by using Image J software ver 1.48 (NIH, USA) [8,9,50]. 

### 4.7. TUNEL Apoptotic Detection

The cells were grown on coverslips after treatments were fixed with 4% paraformaldehyde and permeabilized with 0.25% Triton X-100 in PBS. The cells were then subjected to apoptotic detection using a commercially available TUNEL Alexa Fluor imaging assay kit (Cat. No. C10245, ThermoFisher Scientific) according to the manufacturer’s instructions.

### 4.8. Statistical Analysis

Statistical analyses were performed using the SPSS software version 22.0 (IBM Corp, Armonk, NY, USA). IHC scoring data were compared among groups via a Kruskal–Wallis test, while the difference in the IHC scores between the two ethnic populations at different TNM stages was analyzed via a Mann–Whitney U test. In vitro data are expressed as means ± standard errors of means (SEMs) and were analyzed using an unpaired Student’s *t*-test. A *p*-value < 0.05 was considered statistically significant.

## Figures and Tables

**Figure 1 ijms-24-01776-f001:**
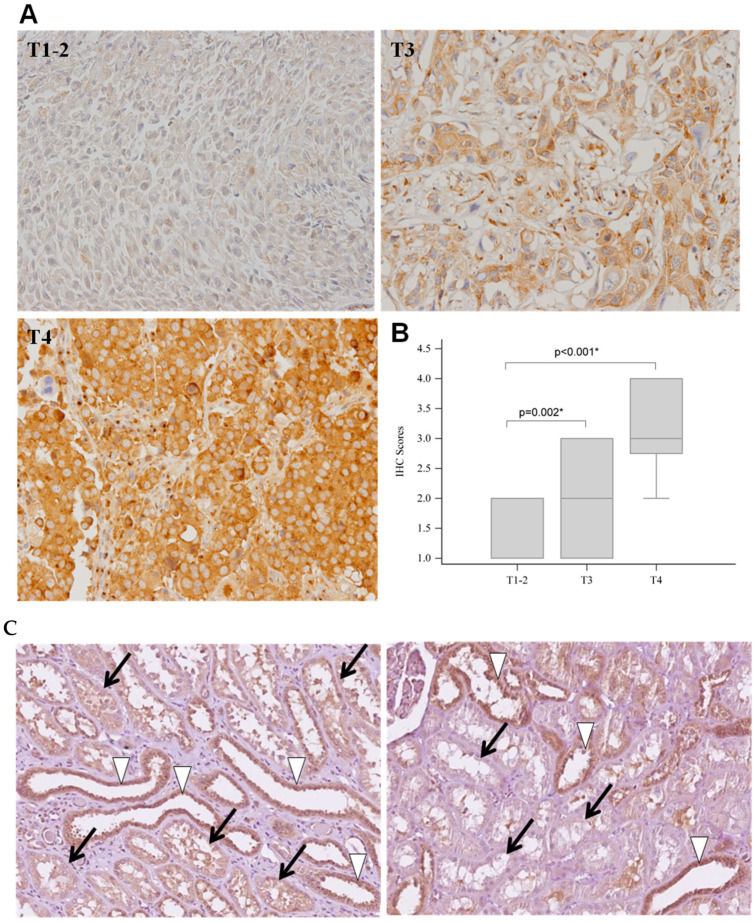
Up-regulation of FGFR3 in renal pelvis tissues from Caucasian patients with urothelial carcinoma. Tissue microarray slides containing sections from Caucasian patients with renal pelvis urothelial carcinoma at stages T1–2 (*n* = 62), T3 (*n* = 17), and T4 (*n* = 5) were subjected to immunohistochemistry (IHC) staining with FGFR3. (**A**) Representative FGFR3-positive stained images from stages T1–2, T3, and T4. (**B**) Quantification of FGFR3 expression levels from IHC staining results (scores were from 1 to 4). Data are shown in box plots displaying 25% to 75% percentile. * indicates significance between the indicated groups. (**C**) Heterogeneous distribution of FGFR3 in normal renal tubular epithelium. Note that proximal renal tubules (black arrows) usually show weak FGFR3 expression (score 1+), while distal tubules (white arrowheads) in the same specimen likely display stronger FGFR3 signals (score 2+~3+).

**Figure 2 ijms-24-01776-f002:**
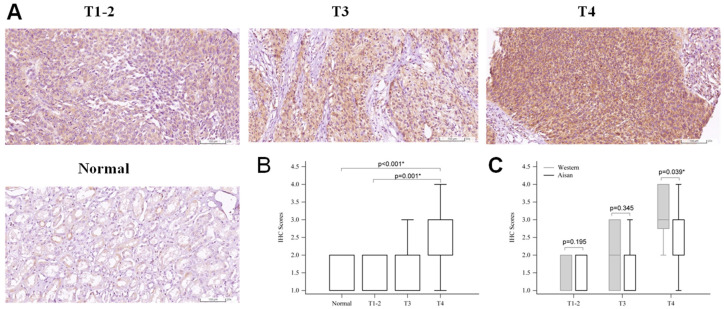
Up-regulation of FGFR3 in renal pelvis tissues from Asian patients with upper tract urothelial carcinoma (UTUC). Renal pelvis tissues were collected from Asian patients diagnosed with UTUC at different TNM stages, including T1–2 (*n* = 44), T3 (*n* = 38), and T4 (*n* = 30). The formalin-fixed paraffin-embedded sections were subjected to immunohistochemistry (IHC) analysis of FGFR3 expression. (**A**) Representative FGFR3-positive stained images from stages T1–2, T3, and T4 and that of archived normal kidney pelvis tissues. (**B**) Analysis of FGFR3 IHC scoring results. (**C**) Statistical analysis of ethnic difference in FGFR3 expression at different UTUC stages. Data are shown in box plots displaying 25% to 75% percentiles. * indicates significance between the indicated groups.

**Figure 3 ijms-24-01776-f003:**
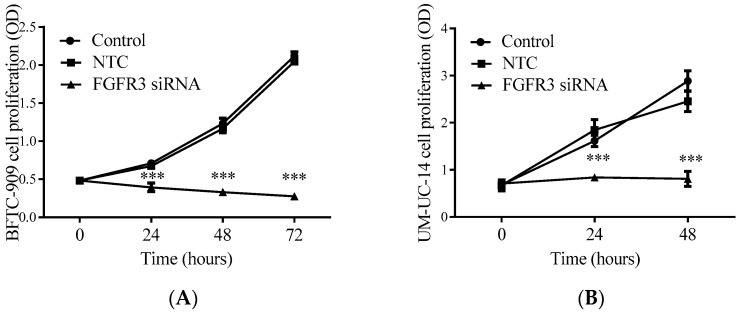
Suppressive effect of *FGFR3* gene silencing on proliferation of cultured UTUC cells. Two UTUC cell lines, BFTC-909 (**A**) and UM-UC-14 (**B**), were treated with FGFR3-targeted siRNA or scramble non-target control (NTC), followed by a WST-1-based proliferation assay. Optical density (OD) measured is expressed as mean ± SEM (*n* = 3). *** *p* < 0.001 compared with respective NTC groups.

**Figure 4 ijms-24-01776-f004:**
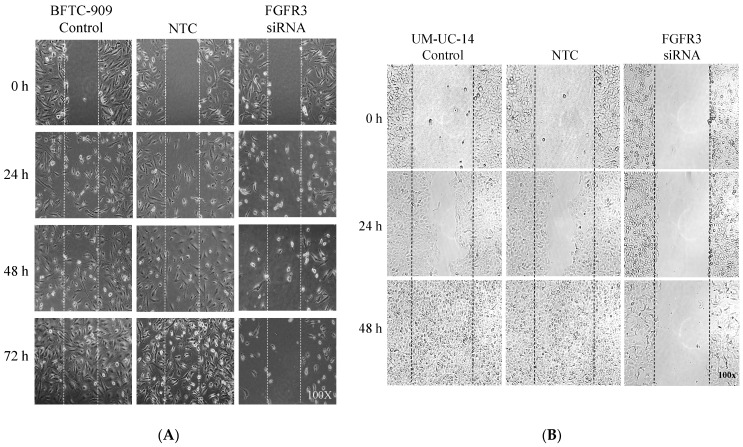
Suppressive effect of *FGFR3* gene silencing on migration of cultured UTUC cells. Two UTUC cell lines, BFTC-909 (**A**) and UM-UC-14 (**B**), were treated with FGFR3-targeted siRNA or scramble non-target control (NTC), followed by consecutive microscopic documentation for the post-wounding closure at the indicated time points.

**Figure 5 ijms-24-01776-f005:**
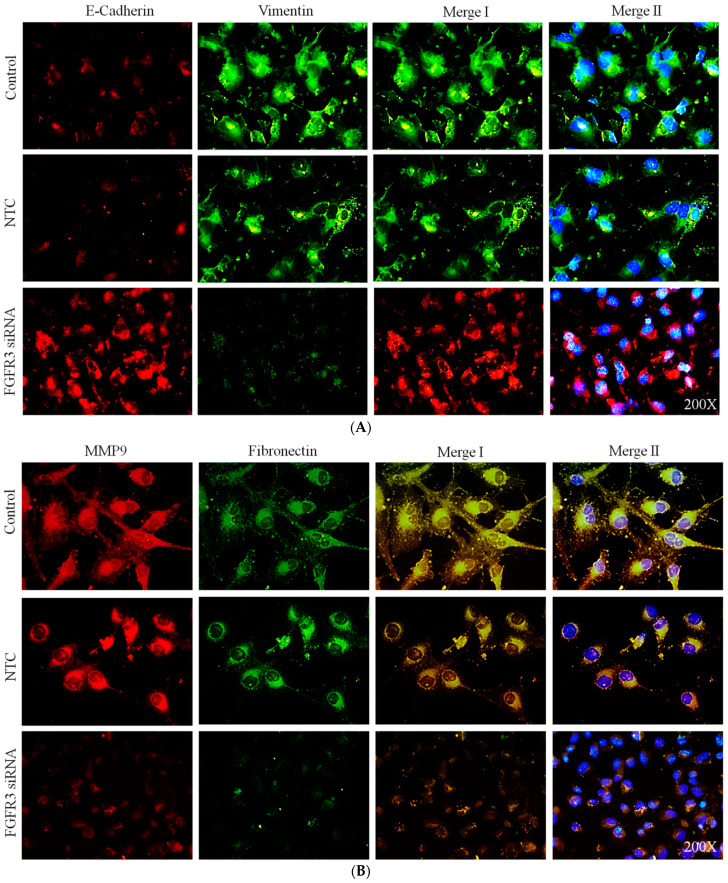
*FGFR3* gene silencing inhibited expression of epithelial–mesenchymal transition markers and transcription factor in cultured UTUC cells. The BFTC-909 cells transfected with FGFR3-targeted siRNA or scramble non-target control (NTC) for 48 h were subjected to immunofluorescent staining for cellular expression of (**A**) E-cadherin (red) and vimentin (green); (**B**) alternatively, for that of MMP9 (red) and fibronectin (green); (**C**) or that of Snail (red) and Claudin-5 (green). Cell nuclei were visualized via DAPI (blue). Merge I images contain signals of red and green channels, while merge II images contain those of triple colors. (**D**) The fluorescence intensities of each channel were quantified by counting 5–10 different fields per sample. Data are expressed as means ± SEM (*n* = 3). * indicates that *p* < 0.05 between the indicated groups. Original magnification: 200×.

**Figure 6 ijms-24-01776-f006:**
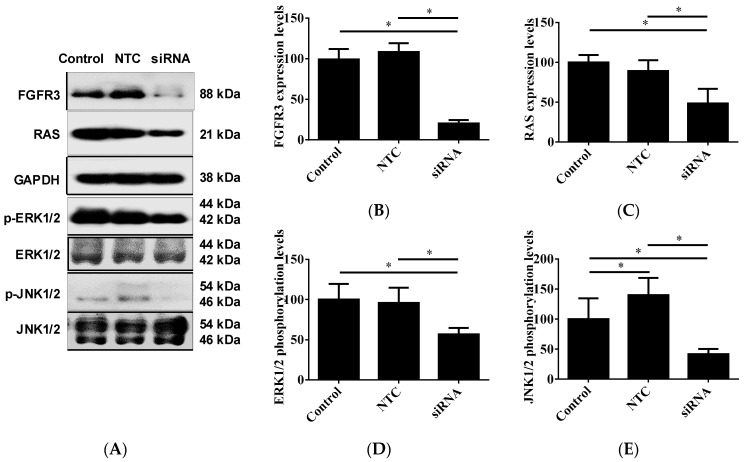
Effect of *FGFR3* gene silencing on constitutive expression of RAS and phosphorylation of MAPK signaling mediators in cultured UTUC cells. (**A**) BFTC-909 cells were treated with FGFR3-targeted siRNA or scramble non-target control (NTC) for 48 h and subjected to Western blotting detection. GAPDH was used as loading internal control. (**B**) Densitometry analyses for protein expression levels of FGFR3 (**B**) and RAS (**C**) are shown as ratios to GAPDH, and phosphorylation levels of EKR1/2 (**D**) and JNK1/2 (**E**) are shown as ratios to respective total proteins. Density data are expressed as mean ± SEM (*n* = 3). * indicates a *p* < 0.05 between the indicated groups.

**Figure 7 ijms-24-01776-f007:**
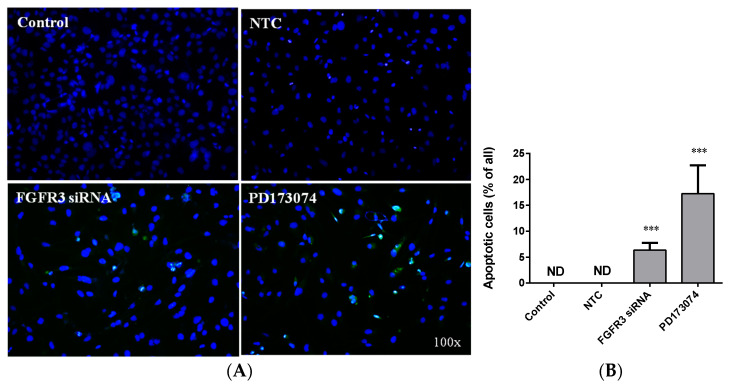
Effects of *FGFR3* gene silencing and FGFR3 kinase inhibition on apoptosis of cultured UTUC cells. (**A**) BFTC-909 cells were treated with either FGFR3-targeted siRNA, scramble non-target control (NTC), or PD173074 at 10 nM for 48 h and subjected to TUNEL fluorescent staining. The TUNEL-positive apoptotic cells show green fluorescence, while DAPI blue fluorescence indicates nuclei of all cells. Original magnification: 100×. (**B**) Morphometry analysis on TUNEL-positive rates in the treated cells. Density data are expressed as mean ± SEM (*n* = 8). *** indicates *p* < 0.001 vs. control group. ND, not detectable.

**Figure 8 ijms-24-01776-f008:**
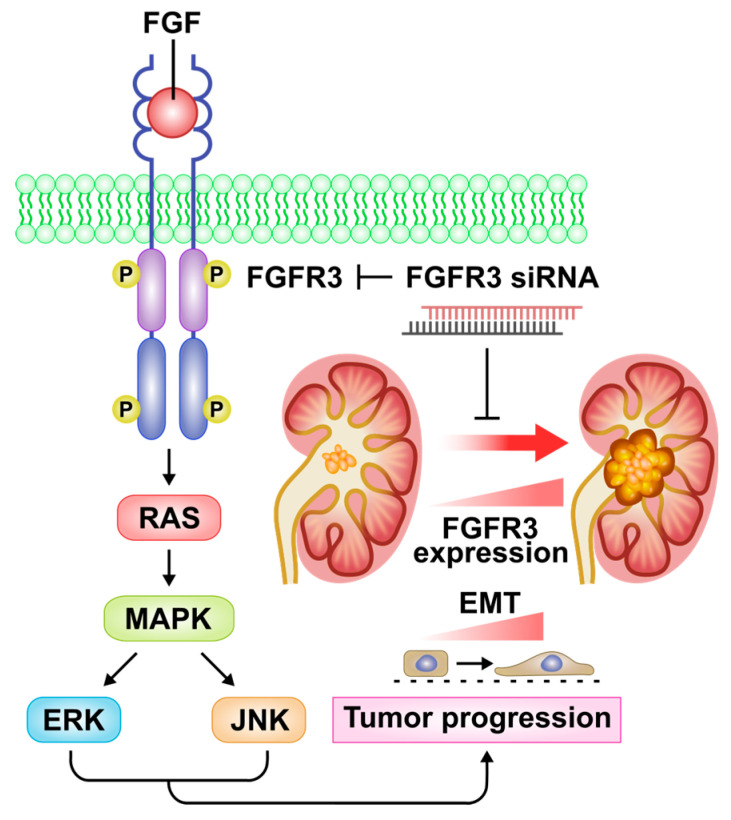
A mechanistic scheme depicting the effects of *FGFR3* gene silencing on RAS/mitogen-activated protein kinase (MAPK) signaling and renal pelvis tumorigenesis. Upon stimulation by fibroblast growth factor (FGF), the intracellular tyrosine kinase domain of FGF receptor 3 (FGFR3) may deliver the mitogenic signal through RAS/MAPK cascade and downstream mediator activation, including extracellular signal-regulated kinase (ERK) and c-Jun N-terminal kinase (JNK). Hence, FGFR3 overexpression and activation may enhance expression of epithelial–mesenchymal transition (EMT) transcription factors and mediators, which eventually contribute to tumor progression. Our findings suggest that FGFR3 siRNA delivery is a useful strategy for alleviating the RAS/MAPK signaling axis and EMT marker expression in renal pelvis urothelial cells, thereby suppressing their metastatic activity and tumor progression.

**Table 1 ijms-24-01776-t001:** FGFR3 positivity rates in renal pelvis tissues from Caucasian patients with upper tract urothelial carcinoma (UTUC).

IHC Scores		TNM Stages	
	T1–T2 (*n* = 62) No. (% Positive)	T3 (*n* = 17) No. (% Positive)	T4 (*n* = 5) No. (% Positive)
0	13 (21%) ^1^	0 (0%)	0 (0%)
1+	36 (58%)	5 (29.4%)	0 (0%)
2+	13 (21%)	7 (41.2%)	1 (20%)
3+	0 (0%)	5 (29.4%)	2 (40%)
4+	0 (0%)	0 (0%)	2 (40%)

^1^ FGFR3 expression was detected via immunohistochemistry (IHC) staining on renal pelvis tissue microarray sections collected from Caucasian patients with UTUC, followed by microscopic scoring.

**Table 2 ijms-24-01776-t002:** FGFR3 positivity rates in renal pelvis tissues from Asian patients with upper tract urothelial carcinoma (UTUC).

IHC Scores	Normal Control (*n* = 50)		TNM Stages	
		T1–T2 (*n* = 44) No. (% Positive)	T3 (*n* = 38) No. (% Positive)	T4 (*n* = 30) No. (% Positive)
0	14 (28%) ^1^	9 (20.5%)	6 (15.8%)	3 (10%)
1+	23 (46%)	21 (47.7%)	13 (34.2%)	6 (20%)
2+	13 (26%)	14 (31.8%)	6 (15.8%)	11 (36.6%)
3+	0 (0%)	0 (0%)	13 (34.2%)	8 (26.7%)
4+	0 (0%)	0 (0%)	0 (0%)	2 (6.7%)

^1^ FGFR3 expression in renal pelvis tissues from patients with UTUC at Kaohsiung Chang Gung Memorial Hospital, Taiwan, was detected via immunohistochemistry (IHC) staining and subsequent microscopic scoring.

## Data Availability

The data presented in this study are available on request from the corresponding authors.

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
