# Peer review of "Genetic Interference of FGFR3 Impedes Invasion of Upper Tract Urothelial Carcinoma Cells by Alleviating RAS/MAPK Signal Activity"

_ijms, 2023, doi:10.3390/ijms24021776_

Round 1

Reviewer 1 Report

Comments:

The manuscript is interesting but requires modifications before publishing.

1.     Based on your study results – “FGFR3 expression increased along with TNM stages in both Asian and Caucasian UTUC tumors despite no statistical difference between two groups”, what is your explanation regarding the stark difference in the prevalence between Asian and Caucasian populations?

2.     Novelty of your study needs to be emphasized.

3.     Limitation of your study needs to be discussed. Right now this is not mentioned at all.

4.     Grammar-related errors need to be rectified.

5.     Spell out the abbreviations when they appear the first time, e.g. RAS/MAPK.

Author Response

Reviewer#1 Comments and Suggestions for Authors

Comments:

The manuscript is interesting but requires modifications before publishing.

The authors’ response:

We appreciate reviewers’ positive and constructive comments on our manuscript. According to your suggestions, we have tried our best to revise the data and text, and highlighted the revised parts in red in the resubmitted manuscript. Below are our "point-to-point" responses to the reviewers’ comments. We would be glad to respond to any further questions and comments, and look forward to hearing from you at your earliest convenience.

  1. Based on your study results – “FGFR3 expression increased along with TNM stages in both Asian and Caucasian UTUC tumors despite no statistical difference between two groups”, what is your explanation regarding the stark difference in the prevalence between Asian and Caucasian populations?

The authors’ response:

We thank reviewer for addressing this issue. Unfortunately, the reason underlying the stark difference in the prevalence between Asian and Caucasian populations cannot be answered by the findings of this study. As originally stated in the Introduction section, “the higher prevalence in Asian populations of UTUC implicates that unique etiologic and genomic factors, distinct from bladder cancer, may contribute to UTUC tumorigenesis [5]”. In this regard, we propose that mutational differences of FGFR3 gene between Asian and Caucasian populations may contribute to this phenomenon. We accordingly added this statement in the Discussion text (in 2nd paragraph).

  1. Novelty of your study needs to be emphasized.

The authors’ response:

We thank reviewer for excellent suggestion. We have rephrased some descriptive sentences in the Discussion text to emphasize the study novelty.

  1. Limitation of your study needs to be discussed. Right now this is not mentioned at all.

The authors’ response:

As suggested by reviewer, we have added additional sentences in the Discussion text to state the study limitation.

  1. Grammar-related errors need to be rectified.

The authors’ response:

We thank reviewer for kindly suggestion. The manuscript text has been polished by MDPI professional English editing.

  1. Spell out the abbreviations when they appear the first time, e.g. RAS/MAPK.

The authors’ response:

We thank reviewer for kindly mentioning. We have thoroughly checked the abbreviations and revised them in the text. In the case of RAS, RAS is a full name but not an abbreviated name.

Reviewer 2 Report

In a paper submitted to MDPI “Cancers” titled: “Genetic interference of FGFR3 impedes invasion of upper tract urothelial carcinoma cells by alleviating RAS/MAPK signal activity” by Gong-Kai Huang et al., authors present experimental data on FGFR-3 supporting role in urothelial cancer. Using IHC staining of human urothelial cancer (UC), FGFR-3 protein staining was histologically assessed, graded and related to ethnicity and stage of cancer, with highest grade (high FGFR-3 expression) observed in Caucasians with advanced stage T3-T4 of UC, and in Asians with FGFR-3 highest FGFR-3 expression in normal epithelium and at early stage T1-T2 of UC development.

In addition, FGFR-3 gene was downregulated by specific siRNA in two urothelial cell lines in vitro, what significantly decreased cancer cells growth in vitro and decreased expression of mesenchymal markers, such as Vimentin, as well as decreased MMP9 protein expression.

Although the paper is providing clear experimental evidence for FGFR-3 involvement in cancer cell survival and aggressive phenotype, the paper lacks additional supporting experiments and couple of controls and therefore is not recommended for publication, but accepted only after major revisions.

The following points should be addressed:

1.     IHC assessment of FGFR-3 protein expression in Table 1 and 2, uses inconsistent number of samples for Asians and Caucasians, with total  112 samples for Asians and 84 for Caucasians. Grade 4 (T4) tumor was represented by just 5 samples for Caucasians and 30 samples for Asian population. In order to derive correlation between the grade of the tumor, FGFR-3 protein expression and ethnic population, the number of patients in each population and the tumor groups should be comparable (e.g., T4 grade tumor 30 different patients in Asians and Caucasians).

2.     FGFR-3 specific siRNA decreases cell growth, please add data supporting specificity of the siRNA sequence, using either FGFR-3 expressing cells that are in-sensitive to downregulation of FGFR-3, or with overexpression of FGFR-3 in cells after first downregulating FGFR-3 with siRNA in a rescue experiment, changes in phenotype.

3.     The siRNA experiments lack cell death assessment, since there is a pronounced effect on cell growth the question remains, if there is induction of apoptosis by FGFR-3 siRNA? Simple Annexin V assay, or any other apoptosis assay would answer this question.

4.     In addition to genetic downregulation of FGFR-3, it is recommended to use small molecule pharmacological inhibitor, such as PD 173074, and compare cell phenotype after inhibitor treatment with growth inhibition induced by FGFR-3 specific siRNA

5.     siRNA in definition is transient, and not stably expressed. For stably transfected urothelial cancer cells, a vector expressing short hairpin RNA (e.g., lentivirus) is transfected and cells are cultured in presence of selective antibiotic to select for stably expressing cells. It is incorrect to introduce siRNA oligo transfection as stable transfection, as it is written in Materials and Methods section: 4.2. Cell culture and FGFR3 siRNA delivery.

6.     In the Western Blot it is recommended to compare P-ERK to a total ERK protein expression and the same with P-JNK to a total JNK kinase, not to a tubulin used as a loading control (Figure 6).

7.     Paper would benefit from careful edits and language correction.

Author Response

Reviewer#2 Comments and Suggestions for Authors

In a paper submitted to MDPI “Cancers” titled: “Genetic interference of FGFR3 impedes invasion of upper tract urothelial carcinoma cells by alleviating RAS/MAPK signal activity” by Gong-Kai Huang et al., authors present experimental data on FGFR-3 supporting role in urothelial cancer. Using IHC staining of human urothelial cancer (UC), FGFR-3 protein staining was histologically assessed, graded and related to ethnicity and stage of cancer, with highest grade (high FGFR-3 expression) observed in Caucasians with advanced stage T3-T4 of UC, and in Asians with FGFR-3 highest FGFR-3 expression in normal epithelium and at early stage T1-T2 of UC development.

In addition, FGFR-3 gene was downregulated by specific siRNA in two urothelial cell lines in vitro, what significantly decreased cancer cells growth in vitro and decreased expression of mesenchymal markers, such as Vimentin, as well as decreased MMP9 protein expression.

Although the paper is providing clear experimental evidence for FGFR-3 involvement in cancer cell survival and aggressive phenotype, the paper lacks additional supporting experiments and couple of controls and therefore is not recommended for publication, but accepted only after major revisions.

The following points should be addressed:

  1. IHC assessment of FGFR-3 protein expression in Table 1 and 2, uses inconsistent number of samples for Asians and Caucasians, with total 112 samples for Asians and 84 for Caucasians. Grade 4 (T4) tumor was represented by just 5 samples for Caucasians and 30 samples for Asian population. In order to derive correlation between the grade of the tumor, FGFR-3 protein expression and ethnic population, the number of patients in each population and the tumor groups should be comparable (e.g., T4 grade tumor 30 different patients in Asians and Caucasians).

The authors’ response:

We thank reviewer for addressing the inconsistent sample numbers between these two ethnic groups. The limited sample number (only 5) of grade 4 (T4) tumors in Caucasian population is mainly due to the fact that we purchased tissue microarray specimens from US Biomax, Inc. (Kit no. HRaU-Uro120Sur-01, Rockville, MD, USA). In this regard, the lower prevalence of UTUC in Caucasians in part limited the available case number. In fact, it is inevitably difficult for Asian researchers to collect additional Caucasian tissue samples. We completely agree that this is indeed the limitation of this study, and we had added the statement in the Discussion text.

  1. FGFR-3 specific siRNA decreases cell growth, please add data supporting specificity of the siRNA sequence, using either FGFR-3 expressing cells that are in-sensitive to downregulation of FGFR-3, or with overexpression of FGFR-3 in cells after first downregulating FGFR-3 with siRNA in a rescue experiment, changes in phenotype.

The authors’ response:

We thank for reviewer’s questioning on the specificity of FGFR3 siRNA. In fact, the western blotting data in Fig. 6A had clearly shown that FGFR3 siRNA delivery exhibited striking knockdown effect on FGFR3 expression in the treated cells, which may support the specificity of siRNA sequence.

  1. The siRNA experiments lack cell death assessment, since there is a pronounced effect on cell growth the question remains, if there is induction of apoptosis by FGFR-3 siRNA? Simple Annexin V assay, or any other apoptosis assay would answer this question.

The authors’ response:

As insightfully suggested by reviewer, we have performed an additional TUNEL staining on the apoptotic cells with FGFR3 siRNA delivery (data shown in the revised Figure 7).

  1. In addition to genetic downregulation of FGFR-3, it is recommended to use small molecule pharmacological inhibitor, such as PD 173074, and compare cell phenotype after inhibitor treatment with growth inhibition induced by FGFR-3 specific siRNA growth inhibition!!

The authors’ response:

We thank reviewer for his/her thoughtful suggestion. As suggested by reviewer, we have performed an additional TUNEL staining on the cells receiving small treatment with molecular inhibitor PD173074 and compared with those receiving FGFR3 siRNA delivery (data shown in the revised Figure 7). The cell morphology of the cells treated with siRNA and PD173074 is presented in the Supplementary Figure S1. The text in the Results and Methodology sections are added accordingly.

  1. siRNA in definition is transient, and not stably expressed. For stably transfected urothelial cancer cells, a vector expressing short hairpin RNA (e.g., lentivirus) is transfected and cells are cultured in presence of selective antibiotic to select for stably expressing cells. It is incorrect to introduce siRNA oligo transfection as stable transfection, as it is written in Materials and Methods section: 4.2. Cell culture and FGFR3 siRNA delivery.

The authors’ response:

We thank reviewer for pinpointing this error and have revised the text “…stably transfected…” to  “…transiently transfected…” in the Materials and Methods section 4.2.

  1. In the Western Blot it is recommended to compare P-ERK to a total ERK protein expression and the same with P-JNK to a total JNK kinase, not to a tubulin used as a loading control (Figure 6).

The authors’ response:

As suggested by reviewer, we have added the western blotting images of total ERK and JNK and revised the densitometrical analysis in the Figure 6.

  1. Paper would benefit from careful edits and language correction.

The authors’ response:

We thank the reviewer's suggestion. The manuscript text has been revised by professional language editing.

Round 2

Reviewer 2 Report

Authors addressed major critiques therefore the manuscript is recommended for publication. Please carefully check minor language misspellings after final edits introduced in the manuscript and Figures outline.

Author Response

Authors' response:

Following reviewer’s kindly suggestion, we have thoroughly checked the manuscript text and corrected one mistyped word (Ras changed to RAS) and rearranged the full name of MAPK in the legend of Figure 8. We additionally mention the revised manuscript text previously underwent language-edited by MDPI.